# Identifying the prevalence and correlates of multimorbidity in middle-aged men and women: a cross-sectional population-based study in four African countries

Lisa K Micklesfield [ID],[1] Richard Munthali,[1] Godfred Agongo [ID],[2,3] Gershim Asiki,[4] Palwende Boua [ID],[5,6] Solomon SR Choma [ID],[7] Nigel J Crowther,[8] June Fabian,[9] Francesc Xavier Gómez-Olivé,[10] Chodziwadziwa Kabudula,[10] Eric Maimela [ID],[7] Shukri F Mohamed [ID],[11] Engelbert A Nonterah,[3,12] Frederick J Raal,[13] Hermann Sorgho,[14] Furahini D Tluway,[15] Alisha N Wade [ID],[10] Shane A Norris [ID],[1,16] Michele Ramsay,[15] as members of AWI-Gen and the H3Africa Consortium

For numbered affiliations see end of article.

**Correspondence to**
Professor Lisa K Micklesfield;
Lisa.Micklesfield@wits.ac.za

## ABSTRACT

**Objectives** To determine the prevalence of multimorbidity, to identify which chronic conditions cluster together and to identify factors associated with a greater risk for multimorbidity in sub-Saharan Africa (SSA).

**Design** Cross-sectional, multicentre, population-based study.

**Setting** Six urban and rural communities in four sub-Saharan African countries.

**Participants** Men (n=4808) and women (n=5892) between the ages of 40 and 60 years from the AWI-Gen study.

**Measures** Sociodemographic and anthropometric data, and multimorbidity as defined by the presence of two or more of the following conditions: HIV infection, cardiovascular disease, chronic kidney disease, asthma, diabetes, dyslipidaemia, hypertension.

**Results** Multimorbidity prevalence was higher in women compared with men (47.2% vs 35%), and higher in South African men and women compared with their East and West African counterparts. The most common disease combination at all sites was dyslipidaemia and hypertension, with this combination being more prevalent in South African women than any single disease (25% vs 21.6%). Age and body mass index were associated with a higher risk of multimorbidity in men and women; however, lifestyle correlates such as smoking and physical activity were different between the sexes.

**Conclusions** The high prevalence of multimorbidity in middle-aged adults in SSA is of concern, with women currently at higher risk. This prevalence is expected to increase in men, as well as in the East and West African region with the ongoing epidemiological transition. Identifying common disease clusters and correlates of multimorbidity is critical to providing effective interventions.

## INTRODUCTION

Sub-Saharan Africa (SSA) is experiencing the highest rate of urbanisation globally and

### STRENGTHS AND LIMITATIONS OF THIS STUDY

⇒ A strength of the study is that it provides multimorbidity prevalence data on a large (n=10 700), population-based sample of middle-aged men and women from sub-Saharan Africa.
⇒ Study sites include urban and rural communities from South, East and West Africa.
⇒ A limitation is the cross-sectional study design with some of the diseases measured via self-report with no longitudinal follow-up.
⇒ The cross-sectional study design only allows for the identification of correlates of multimorbidity, rather than causal factors.

together with an increase in life expectancy must prepare for the prevalence of noncommunicable diseases (NCDs) to increase further.[1] Multimorbidity, the co-occurrence of two or more chronic diseases in one individual, is common, challenging the affected individual, their attending healthcare professionals and an overstretched health system. A recent systematic review highlighted a paucity of multimorbidity costing studies from low-income and middle-income countries (LMICs).[2] When describing global patterns of multimorbidity, Afshar *et al* reported a positive but non-linear association between country GDP and multimorbidity prevalence, and also identified an inverse association between multimorbidity and socioeconomic status (SES) in countries with the highest GDP, with the gradient sometimes reversed in countries with lower GDP.[3] Understanding the social and structural forces that drive the clustering of diseases, thereby exacerbating

their impact on disease outcomes, is defined as syndemics[4] and is critical to our understanding of multimorbidity.

Recent data from LMICs, including South Africa, have been reported in a scoping review of NCD multimorbidity, which ranged in prevalence from 3.2% to 90.5%, and reported that 95.3% of the studies found female sex to be a risk factor for multimorbidity.[5] The studies included in this review used various chronic diseases comprising the multimorbidity profile, and different diagnostic criteria for each chronic disease, highlighting the need for a consensus on the definition of multimorbidity and the core conditions that it is comprised of. Further, the inclusion of chronic infectious diseases such as HIV and tuberculosis should also be considered when quantifying the burden of multimorbidity but may be more relevant in some settings than others. There is a dearth of multimorbidity prevalence data from Africa, with studies reporting data from single countries such as South Africa,[6 7] Ghana,[8] Burkina Faso[9] and Kenya.[10] Although the influence of age and SES on multimorbidity prevalence is well recognised globally,[3 5 11 12] several African studies have also identified associations between lifestyle factors, mental health and multimorbidity.[7 10 12] Comparing and contrasting different African settings and identifying which diseases cluster together and the factors associated with this clustering will assist in the design of effective interventions.

The double burden of non-communicable and infectious diseases as well as the impact of urbanisation on disease risk highlights the need to undertake multimorbidity research in Africa, and the heterogeneity in the epidemiological and nutrition transition, and disease burden of different African countries, provides an opportunity to compare and contrast the correlates and prevalence of multimorbidity across different African settings. The aim of this study was to determine the prevalence of multimorbidity, to identify which chronic conditions cluster together and to identify factors associated with a greater risk for multimorbidity in six rural and urban communities in four SSA countries. This study will contribute to the evidence base on multimorbidity by providing prevalence data across different African settings and identifying common disease clusters to potentially understand the pathogenesis of different diseases. This will inform the design of more effective interventions and provide formative data on multimorbidity for policy-makers in the planning and implementation of more effective health systems with particular relevance to other LMICs.

## METHODS
### Study population
Data included in this study are from the Africa Wits-INDEPTH (University of the Witwatersrand, Johannesburg, and the International Network for the Demographic Evaluation of Populations and Their Health) partnership for Genomic Studies (AWI-Gen),[13] which is a National Institutes of Health-funded Collaborative Centre of the Human Heredity and Health in Africa (H3Africa) Consortium. AWI-Gen is a population-based cross-sectional study of adults and includes six participating urban and rural centres in four SSA countries. West Africa included two countries, Ghana (Navrongo) and Burkina Faso (Nanoro); East Africa included Kenya (Nairobi); and South Africa had three study sites (Soweto, Agincourt and Dikgale). The details of the recruitment strategy as well as other data collected are described in Ali *et al.*[14] For this study, we only included participants from 40 to 60 years (n=10 700).

### Sociodemographic and anthropometric data
Standard structured AWI-Gen questionnaires were administered by trained field staff with some country and site-specific modifications to suit their context.[14] Data on self-reported sociodemographics (age, highest level of education attained, partnership status, employment status) were collected. Each study participant was assigned to an SES quintile, which was determined for each study site by categorising factor scores that were predicted from a principal component analysis of the number of household assets. Alcohol consumption was categorised as never, current problematic, current non-problematic or former consumer,[15] but was not available for the Soweto women. Smoking of tobacco products was categorised as never, current or previous smoker. Total moderate–vigorous intensity physical activity (MVPA) was calculated as minutes per week from the accumulation of occupation, walking for travel and leisure time activity collected by self-report using the Global Physical Activity Questionnaire.[16] Weight and height were used to calculate body mass index (BMI: weight $(kg)$/height $(m^2)$).

### Multimorbidity
We defined multimorbidity as the presence of two or more of the following seven conditions (HIV infection, cardiovascular disease (CVD), chronic kidney disease (CKD), asthma, diabetes, dyslipidaemia and hypertension) for all sites except for women from Soweto, where data was only available for four of the conditions (HIV infection, diabetes, dyslipidaemia and hypertension). Obesity was not included as one of the conditions as in the context of this paper it is considered an NCD risk factor rather than a disease. For this reason, BMI was included as an independent variable in the multinomial regression.

HIV status was self-reported although participants in South Africa and Kenya were offered a voluntary government-approved rapid HIV test. Due to the low prevalence of HIV in Burkina Faso and Ghana, participants were not offered HIV tests.

Hypertension was defined as systolic blood pressure≥140 mm Hg and/or diastolic blood pressure≥90 mm Hg, in line with the seventh report of the Joint National Committee on Prevention, Detection, Evaluation, and Treatment of High Blood Pressure, or if the participant was taking hypertension medication.[17]

Diabetes was defined using the WHO criteria, which are the presence of a previous diagnosis of diabetes by a healthcare professional or fasting blood glucose≥7 mmol/L or

random glucose≥11.1 mmol/L, or on diabetes medication at the time of recruitment.[18]

Dyslipidaemia was defined as the presence of one of the following: total cholesterol (TC)≥5.0 mmol/L, or high-density lipoprotein cholesterol (HDL-C)<1.0 mmol/L for men and <1.3 mmol/L for women, or low-density lipoprotein cholesterol (LDL-C) ≥3. 0 mmol/L, or triglycerides (TG)≥1.7 mmol/L, or on lipid-modifying medication,[19] or self-report of ever being diagnosed by a health professional with high cholesterol. The Randox Daytona Plus (Randox Laboratories, UK) autoanalyzer was used to analyse HDL-C, TG and TC on fasting venous blood samples. LDL-C was calculated using the Friedewald equation.[20]

CKD was defined as estimated glomerular filtration rate (eGFR)<60 mL/min per 1.73 m$^2$ (calculated using the Chronic Kidney Disease Epidemiology (CKD-EPI) (creatinine) equation 2009, without adjustment for African American ethnicity), presence of albuminuria (urine albumin creatinine ratio>3 mg/mmol) or both.[21] As the study was cross-sectional, low eGFR and albuminuria were not confirmed with follow-up testing. At each partner site, blood and urine specimens were processed and stored at −112°F. After completion of the study, each partner site transported samples on dry ice to a central laboratory in Johannesburg, South Africa. At the central laboratory, all analyses were batched and performed according to good laboratory practice with external monitoring for quality control. Serum and urine creatinine (mg/dL) were measured using Jaffe's kinetic method calibrated to an isotope dilution mass spectrometry-traceable standard. Urinary albumin concentration was measured with immunoturbidimetry. The urine albumin to urine creatinine ratio (mg/mmol) was calculated from these measurements.

CVD was defined as present if the participant reported having had a heart attack or stroke or transient ischaemic attack. Participants previously diagnosed with congestive heart failure or angina were also classified as having CVD. No data were available for angina or heart attack prevalence for the Soweto men, so CVD prevalence was calculated using the remaining data.

Asthma was determined by self-report or use of medication for the condition.

## Statistical analysis
Data were summarised using means and standard deviations (±SD) for continuous parametric data, and median (IQR) for non-parametric data. A Student's t-test was used to test for differences between men and women on parametric continuous variables within each site, while Kruskal-Wallis test was used for non-parametric data. Chi-square test was used for categorical data. Due to significant sex differences within sites (online supplemental tables 1–4), all further analyses were stratified by sex. Since the outcome had three categories, that is, zero conditions, one condition and at least two conditions (multimorbidity) of the seven conditions under study, multinomial logistic regression was used to explore the factors associated with either one condition or at least two conditions, with none as the reference group, in men and

women separately. A p value<0.05 was considered statistically significant in all tests carried out using STATA V.14.1 SE.

To understand the different multimorbidity patterns between men and women, within and between countries, and between regions, that is, South (South African sites), East (Kenyan site) and West (Ghana and Burkina Faso) Africa, different diseases were plotted using UpSetR, an R package.[22] Separate analyses were done for men and women, separated by site and then geographic region.

### Patient and public involvement
This study did not involve any patients and/or the public.

## RESULTS
### Sociodemographic and lifestyle factors
Only participants between the ages of 40 and 60 years were selected for these analyses (n=10 700), and the median age ranged from 48 years in Nairobi to 51 years in Agincourt, Dikgale and Navrongo (online supplemental tables 1 and 2). The only site where there was a significant sex difference in age was Navrongo where women were older than men. Sex differences in education level attained were significant at all sites, except Dikgale where the majority of participants had completed secondary school (55.2%), while in Navrongo and Nanoro the majority of the participants did not have any formal education. Employment was above 90% in both Nairobi and Nanoro, 62.6% and 60% in Navrongo and Soweto, respectively, and only 36.1% in Agincourt and 37.7% in Dikgale. There were significant sex differences in SES at all six sites. Smoking and alcohol consumption were significantly different between the sexes at all sites except Soweto (alcohol status was not known in the women), as previously reported.[23]

### Multimorbidity prevalence
For the total study population as well as when the sites were combined by geographical area, multimorbidity prevalence was higher in women compared with men, with nearly 50% of the women in the total sample presenting with multimorbidity compared with 35% of the men (figure 1). Multimorbidity prevalence was highest in the South African men and women (51.7% and 64.9%, respectively), followed by East Africa (31.3% and 48.4%, respectively) and then West Africa (20.2% and 24.1%, respectively). Overall, the site with the highest multimorbidity prevalence was Agincourt with 66.6%, and the site with the lowest prevalence was Nanoro with 21.2% (online supplemental tables 3 and 4).

### Multimorbidity clusters
Multimorbidity clustering for the total study population, and for the three geographical regions, stratified by sex, are presented in figures 2 and 3, respectively.

### Study population, stratified by sex
The majority of the total study population of men (42.9%) and women (44.8%) had 1 disease only, and in those

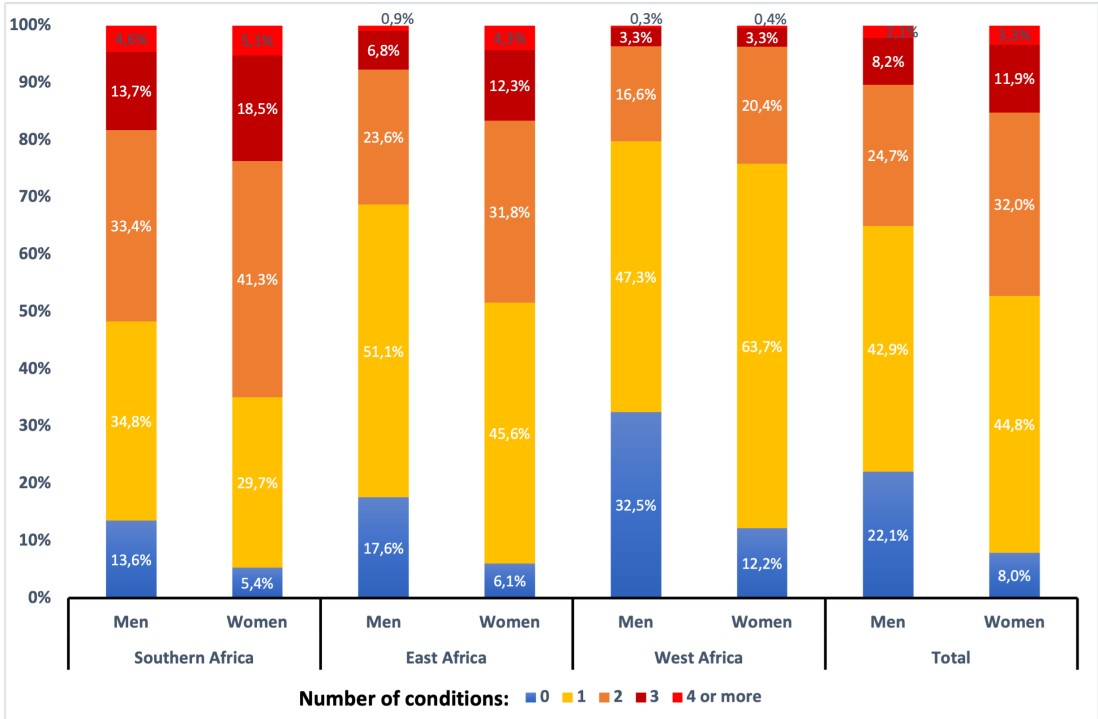

**Figure 1** Prevalence of 0, 1, 2, 3 and 4 or more chronic conditions (including HIV infection, cardiovascular disease, chronic kidney disease, asthma, diabetes, dyslipidaemia and hypertension) in men and women from South, East and West Africa, and the full AWI-Gen cohort.

with only 2 diseases, the most common combination was dyslipidaemia and hypertension (men 12%; women 18%) (figure 2A,B). While in men, the most common combination of three diseases was dyslipidaemia, hypertension and CKD (2% of the total study population), in women the most common combination was dyslipidaemia, hypertension and HIV infection (3.2%). The prevalence of 4 or more diseases was marginally higher in women compared with men (3.3% vs 2.1%) with the most common cluster of 4 diseases being dyslipidaemia, hypertension, HIV and CKD in both sexes.

### Geographical regions, stratified by sex

The majority of men (51.7%) and women (64.9%) living in South Africa presented with multimorbidity (online supplemental table 2, figure 1), with two diseases (hypertension and dyslipidaemia) (figure 3A,B) as the most common combination in the men (33.4%) and women (41.3%). In contrast the majority of men (51.1%) and women (45.6%) from East Africa reported only one disease, although the pattern in those with two diseases was similar to South Africa as dyslipidaemia and hypertension were the most common cluster (figure 3C,D), and this was the same in West Africa (figure 3E,F). It is in the clustering of three or more diseases that there are differences between the sites as the most common clustering of three diseases in South African men and women was dyslipidaemia, hypertension and HIV, while in East and West Africa dyslipidaemia and hypertension were most commonly clustered with CKD.

### Sites, stratified by sex

At all sites except Nanoro, the prevalence of multimorbidity was higher in women than in men, with the prevalence being highest in Agincourt women (74.1%) and lowest in men from Navongo (18%) (online supplemental tables 3 and 4). In all the sites, the most common disease combination in men and women with two diseases only was dyslipidaemia and hypertension (online supplemental figure 1–12), with the lowest prevalence in Nanoro women (7.6%) and the highest prevalence in Soweto women (30.5%). There were differences between the sites in the most common disease clusters in men and women with three diseases only, with dyslipidaemia and hypertension clustered with CKD in Soweto men, Nairobi men and women, Navrongo men and women and Nanoro women. In men from Dikgale, and Agincourt men and women, the most common disease cluster in those with only three diseases was dyslipidaemia, hypertension and HIV, while in Dikgale and Soweto women, and Nanoro men, it was dyslipidaemia, hypertension and diabetes. The highest prevalence with 4 or more diseases was reported in women from the South African sites: 8% of the Agincourt women and 7.6% of the Dikgale women (data was not available for Soweto women). Except for Nairobi women, <1% of the men and women from the East and West African sites presented with 4 or more diseases.

### Multinomial logistic regression, stratified by sex

Risk factors associated with having one condition and with having multimorbidity, compared with no condition,

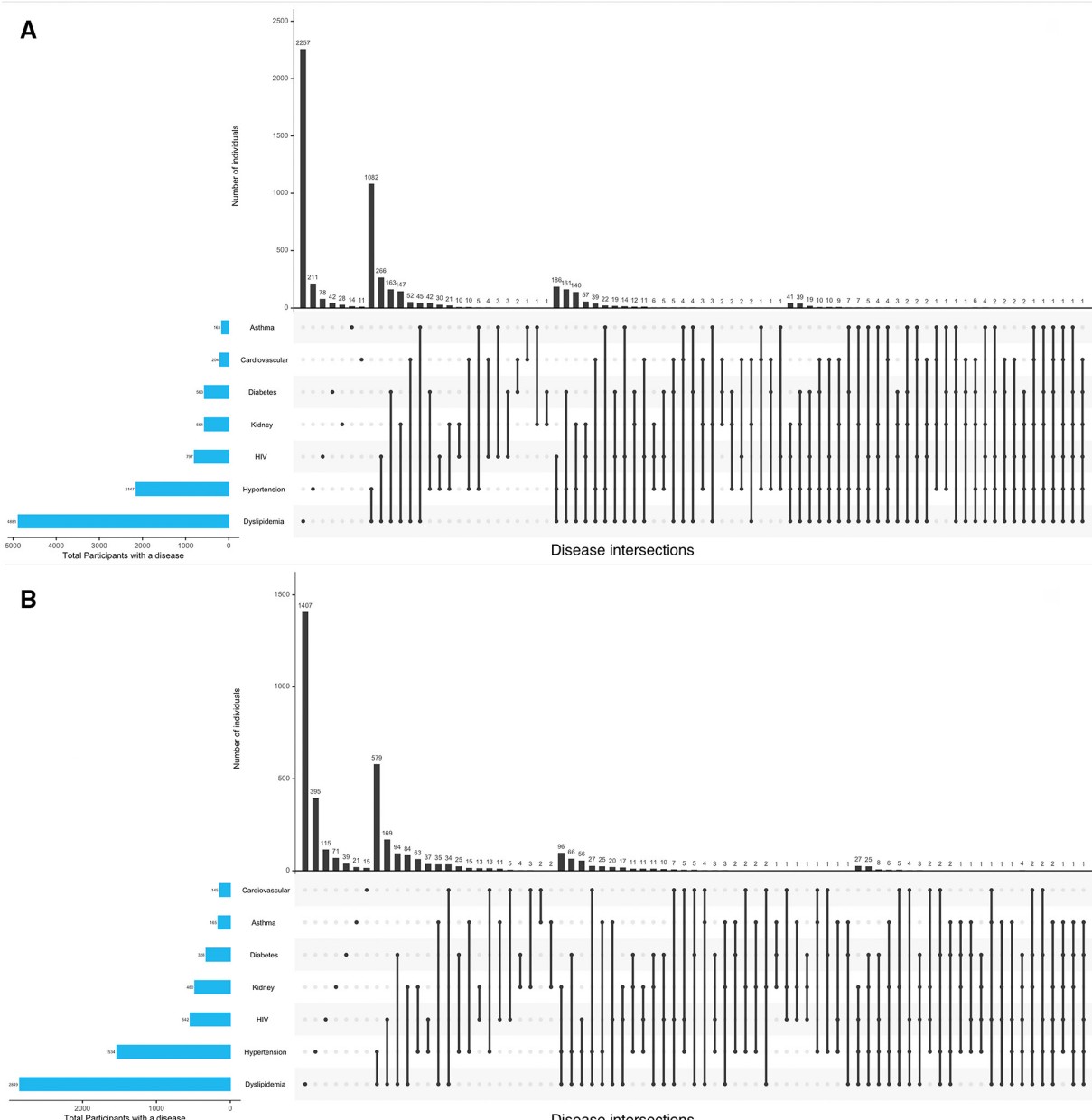

**Figure 2** Multimorbidity clustering for the total AWI-Gen cohort in (A) women and (B) men.

were determined using multinomial regression and are presented for women (table 1) and men (table 2).

Due to the lower prevalence of multimorbidity in women from East and West Africa compared with South Africa, women living in Nairobi and Nanoro were more likely to have one condition compared with Agincourt women, and conversely, together with women from Navrongo, they were at a lower relative risk of multimorbidity compared with women from Agincourt. In women, being older was associated with a higher relative risk of multimorbidity, and a 1 kg/m$^2$ higher BMI was associated with a 7% higher risk of having 1 condition and an 11% higher risk of multimorbidity. Further, being a former consumer of alcohol compared with never consuming alcohol was associated with a higher risk of having one disease, and with multimorbidity. None of the other

measures of SES or lifestyle behaviours were associated with multimorbidity risk in women.

When compared with the Agincourt men, the relative risk of multimorbidity was lower at all sites, except Soweto. Similar to the women, older age and BMI were associated with a higher risk of multimorbidity with every year of age being associated with a 3% higher risk of multimorbidity, and a 1 kg/m$^2$ higher BMI was associated with a 9% higher risk of having 1 condition and a 14% higher risk of multimorbidity. While smoking was not associated with disease risk, the current consumption of alcohol, whether non-problematic or problematic, was associated with a lower risk of having one disease, and current non-problematic consumption of alcohol was associated with a 32% lower risk of multimorbidity. Another lifestyle behaviour that was associated with disease risk was time spent in MVPA,

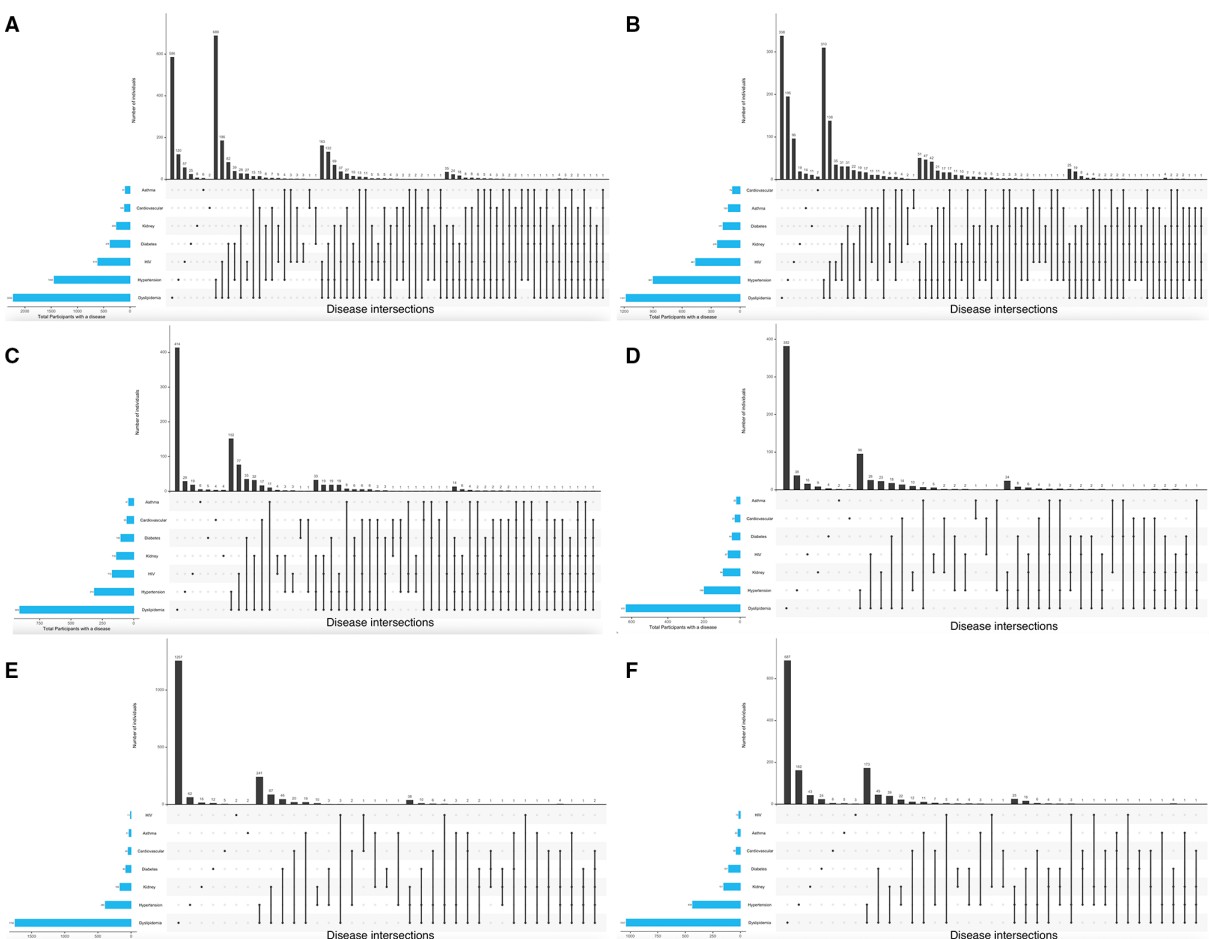

**Figure 3** Multimorbidity clustering in (A) South African women, (B) South African men, (C) East African women, (D) East African men, (E) West African women and (F) West African men.

with more time spent in MVPA associated with a lower risk of having one condition, and with multimorbidity. Living with a partner either presently or in the past was associated with a higher risk of one condition and an even higher risk of multimorbidity when compared with not living with a partner. The attainment of primary school level education was associated with a 30% higher risk of multimorbidity compared with no formal education, and being employed was associated with a 33% lower risk of having multimorbidity compared with being unemployed; however, there was no association between SES quintile and risk for one disease or multimorbidity.

## DISCUSSION

In this study of middle-aged men and women living in South, East and West Africa we have reported sex differences in the prevalence of multimorbidity, and identified socio-demographic, socio-economic and lifestyle factors associated with multimorbidity risk in men and women. While the prevalence of multimorbidity was more than 50% in South Africa, it was only just over 20% in West Africa, with East Africa reporting a prevalence of 31.3% in men and 48.4% in women. What was consistent across all sites was the higher prevalence of multimorbidity

in women compared with men, and in both men and women, age and BMI were independently associated with a higher risk of multimorbidity. These findings highlight the need for longitudinal studies of ageing African populations to examine the effect of multimorbidity on mortality, as studies from high-income countries have shown that this is influenced by disease type, number and certain combinations that comprise multimorbidity.[24 25] Our study has also shown that the most commonly occurring disease cluster at all SSA sites was dyslipidaemia with hypertension.

A recent systematic review of prevalence studies from South Africa reported a 3%–23% prevalence of multimorbidity in studies with a wide range of age groups and 30%–71% in older adults.[26] The higher prevalence of multimorbidity in the South African sites in the current study (54%–66%) may be attributed to the advanced epidemiological transition and increasing urbanisation in South Africa.[27] This coincides with a higher BMI and prevalence of obesity in the three South African sites compared with the West and East African sites.[28] Despite being considered a rural site, Agincourt reported the highest prevalence of multimorbidity at 66.6%. This prevalence is similar to those reported from the same study

**Table 1** Multinomial logistic regression to determine factors associated with either one condition or multimorbidity, with none as the reference group, in AWI-Gen women

| | No disease versus one disease | | | No disease versus multimorbidity | | |
|---|---|---|---|---|---|---|
| | RRR | 95% CI | P value | RRR | 95% CI | P value |
| Site (Ref: Agincourt) | | | | | | |
| Dikgale | 1.42 | 0.84 to 2.43 | 0.185 | 0.77 | 0.46 to 1.28 | 0.306 |
| Nairobi | 1.92 | 1.18 to 3.11 | 0.009 | 0.58 | 0.36 to 0.93 | 0.023 |
| Nanoro | 3.10 | 1.78 to 5.41 | 0.000 | 0.38 | 0.22 to0.67 | 0.001 |
| Navrongo | 1.56 | 0.97 to 2.52 | 0.068 | 0.25 | 0.16 to 0.41 | 0.000 |
| Age (years) | 1.00 | 0.98 to 1.02 | 0.932 | 1.04 | 1.02 to 1.06 | 0.000 |
| BMI (kg/m$^2$) | 1.07 | 1.04 to 1.10 | 0.000 | 1.11 | 1.08 to 1.14 | 0.000 |
| Smoking status (Ref: never smoked) | | | | | | |
| Current smoker | 0.58 | 0.27 to 1.26 | 0.171 | 0.82 | 0.38 to 1.80 | 0.628 |
| Previous smoker | 0.56 | 0.27 to 1.18 | 0.126 | 0.76 | 0.37 to 1.58 | 0.468 |
| Alcohol consumption (Ref: never consumed) | | | | | | |
| Current non-problematic consumer | 0.86 | 0.64 to 1.16 | 0.327 | 0.97 | 0.70 to 1.34 | 0.851 |
| Current consumer (problematic) | 1.01 | 0.66 to 1.55 | 0.959 | 1.14 | 0.72 to 1.79 | 0.580 |
| Former consumer | 1.49 | 1.04 to 2.12 | 0.028 | 1.97 | 1.37 to 2.84 | 0.000 |
| GPAQ MVPA (min/week/100) | 0.98 | 0.99 to 1.01 | 0.550 | 0.99 | 0.99 to 1.00 | 0.132 |
| Partnership status (Ref: never married or cohabited) | | | | | | |
| Married or cohabiting | 1.40 | 0.82 to 2.4 | 0.223 | 1.13 | 0.66 to 1.90 | 0.658 |
| Divorced, separated, partner deceased | 1.62 | 0.93 to 2.82 | 0.090 | 1.48 | 0.86 to 2.54 | 0.157 |
| Highest level of education attained (Ref: no formal education) | | | | | | |
| Primary education | 1.12 | 0.80 to 1.57 | 0.505 | 1.34 | 0.95 to 1.90 | 0.094 |
| Secondary education | 1.31 | 0.84 to 2.03 | 0.235 | 1.51 | 0.97 to 2.36 | 0.067 |
| Tertiary education | 2.95 | 0.67 to 12.9 | 0.152 | 2.81 | 0.65 to 12.18 | 0.168 |
| Employment status (Ref: unemployed) | | | | | | |
| Employed | 0.76 | 0.57 to 1.02 | 0.069 | 0.79 | 0.59 to 1.07 | 0.125 |
| SES quintile (Ref: quintile 1) | | | | | | |
| Quintile 2 | 0.96 | 0.68 to 1.34 | 0.799 | 1.16 | 0.81 to 1.66 | 0.412 |
| Quintile 3 | 1.27 | 0.88 to 1.83 | 0.195 | 1.35 | 0.92 to 1.98 | 0.127 |
| Quintile 4 | 0.99 | 0.70 to 1.39 | 0.945 | 1.03 | 0.72 to 1.48 | 0.857 |
| Quintile 5 | 0.99 | 0.68 to 1.42 | 0.940 | 1.15 | 0.78 to 1.68 | 0.489 |

BMI, body mass index; GPAQ, Global Physical Activity Questionnaire; MVPA, moderate–vigorous intensity physical activity; RRR, relative risk ratio; SES, socioeconomic status.

site but including larger sample sizes of participants older than 40 years (n>3000) and different chronic diseases in the definition of multimorbidity.[29 30] Kabudula *et al*[31] have described a 'protracted' epidemiological transition in Agincourt between 1993 and 2013 as a result of the coexistence of infectious and NCDs, as well as the influence of social changes, and their results highlight the different transition experience in LMICs compared with high-income countries.[27] Although the West African sites, Ghana and Burkina Faso, reported the lowest prevalence of multimorbidity and a similar prevalence between men and women, it is expected that this will increase, following a similar trajectory to the South African sites. Mohamed *et al*[10] have reported a lower multimorbidity prevalence of 28.7% (31.5% in women vs 21.4% in men) in the same AWI-Gen study participants from Nairobi, Kenya. Reasons for the different prevalence compared with the current study is that they included 16 chronic conditions, and all measurements, except for hypertension and obesity, were via self-report. This may have resulted in an underestimation of multimorbidity prevalence as it is well recognised that many of these chronic diseases are undiagnosed in Africa.[32–34] The higher prevalence of multimorbidity in women compared with men has been well described,[35] with the studies from South Africa showing less consistent results.[26]

Dyslipidaemia was the most commonly occurring disease across all the sites, with the prevalence ranging from 42.7% in men from Navrongo to 87.4% in women from Dikgale. These results are similar to other African studies,[36 37] with a recent study by Masilela *et al*,[38] reporting a prevalence of 76.7% in South African adults receiving

**Table 2** Multinomial logistic regression to determine factors associated with either one condition or multimorbidity, with none as the reference group, in AWI-Gen men

| | No disease versus one disease | | | No disease versus multimorbidity | | |
|---|---|---|---|---|---|---|
| | RRR | 95% CI | P value | RRR | 95% CI | P value |
| Site (Ref: Agincourt) | | | | | | |
| Dikgale | 0.75 | 0.48 to 1.17 | 0.204 | 0.59 | 0.38 to 0.92 | 0.019 |
| Nairobi | 0.95 | 0.64 to 1.40 | 0.781 | 0.42 | 0.28 to 0.63 | 0.000 |
| Nanoro | 0.80 | 0.54 to 1.18 | 0.256 | 0.29 | 0.19 to 0.43 | 0.000 |
| Navrongo | 0.55 | 0.38 to 0.80 | 0.002 | 0.16 | 0.11 to 0.24 | 0.000 |
| Soweto | 1.15 | 0.75 to 1.78 | 0.515 | 1.19 | 0.77 to 1.84 | 0.422 |
| Age (years) | 1.01 | 0.99 to 1.02 | 0.369 | 1.03 | 1.02 to 1.05 | 0.000 |
| BMI (kg/m$^2$) | 1.09 | 1.06 to 1.12 | 0.000 | 1.14 | 1.11 to 1.18 | 0.000 |
| Smoking status (Ref: never smoked) | | | | | | |
| Current smoker | 1.07 | 0.87 to 1.32 | 0.520 | 0.95 | 0.75 to 1.20 | 0.671 |
| Previous smoker | 1.25 | 0.99 to 1.58 | 0.061 | 1.17 | 0.91 to 1.52 | 0.223 |
| Alcohol consumption (Ref: never consumed) | | | | | | |
| Current non-problematic consumer | 0.66 | 0.52 to 0.85 | 0.001 | 0.68 | 0.51 to 0.89 | 0.006 |
| Current consumer (problematic) | 0.63 | 0.47 to 0.83 | 0.001 | 0.77 | 0.56 to 1.06 | 0.105 |
| Former consumer | 1.09 | 0.81 to 1.48 | 0.571 | 1.10 | 0.79 to 1.54 | 0.568 |
| GPAQ MVPA (min/week/100) | 0.99 | 0.99 to 0.10 | 0.015 | 0.99 | 0.99 to 0.10 | 0.004 |
| Partnership status (Ref: never married or cohabited) | | | | | | |
| Married or cohabiting | 1.39 | 1.01 to 1.92 | 0.045 | 1.53 | 1.10 to 2.15 | 0.011 |
| Divorced, separated, partner deceased | 1.61 | 1.08 to 2.40 | 0.018 | 2.43 | 1.62 to 3.65 | 0.000 |
| Highest level of education attained (Ref: no formal education) | | | | | | |
| Primary education | 1.11 | 0.89 to 1.40 | 0.355 | 1.30 | 1.00 to 1.69 | 0.049 |
| Secondary education | 1.10 | 0.85 to 1.44 | 0.460 | 1.22 | 0.91 to 1.64 | 0.183 |
| Tertiary education | 1.04 | 0.65 to 1.68 | 0.858 | 1.10 | 0.67 to 1.82 | 0.710 |
| Employment status (Ref: unemployed) | | | | | | |
| Employed | 0.94 | 0.75 to 1.17 | 0.555 | 0.77 | 0.61 to 0.98 | 0.033 |
| SES quintile (Ref: quintile 1) | | | | | | |
| Quintile 2 | 1.09 | 0.83 to 1.43 | 0.525 | 1.18 | 0.87 to 1.61 | 0.293 |
| Quintile 3 | 1.07 | 0.81 to 1.41 | 0.630 | 1.08 | 0.79 to 1.49 | 0.632 |
| Quintile 4 | 1.12 | 0.86 to 1.48 | 0.381 | 1.11 | 0.81 to 1.51 | 0.512 |
| Quintile 5 | 1.03 | 0.78 to 1.36 | 0.823 | 1.303 | 0.96 to 1.78 | 0.093 |

BMI, body mass index; GPAQ, Global Physical Activity Questionnaire; MVPA, moderate–vigorous intensity physical activity; RRR, relative risk ratio; SES, socioeconomic status.

care for diabetes and hypertension, and Reiger *et al*[39] reporting a prevalence of 67.3% in over 4000 adults aged 40 years and older from Agincourt, South Africa. Many of these African studies have reported that low HDL-C is the main driver of the high prevalence of dyslipidaemia, and it is important to debate whether the current international cut-offs for HDL-C, triglyceride and other cholesterol measures are relevant to the African population. A low HDL-C is frequently seen with obesity and the metabolic syndrome and is associated with hypertriglyceridaemia, particularly post prandially.[40] The main driver of atherosclerotic CVD is LDL-C so this may be a more important marker of risk for CVD.[41] If this is the main criterion for defining dyslipidaemia and low HDL-C were excluded from the definition, then the prevalence of

dyslipidaemia would be considered to have a lesser effect in the context of multimorbidity. Data from Agincourt has reported that while low HDL (<1.19 mmol/L) was prevalent in 26.5% of the adults aged 40 years and older, high LDL (>4.1 mmol/L) was measured in only 3.7%.[42]

In the current study, the most common cluster of two diseases at all sites was dyslipidaemia with hypertension, with this prevalence being higher than the prevalence of any single disease in Agincourt women, and Soweto men and women. This is in contrast to a recent systematic review from South Africa which identified the common disease clusters as hypertension and diabetes, hypertension and HIV, and tuberculosis (TB) and HIV.[26] Irrespective of the difference in findings, these studies highlight the high prevalence of hypertension in these

large samples of middle-aged men and women from East, West and South Africa as previously reported by Gómez-Olivé *et al*.[34] A recent systematic review and meta-analysis assessing CKD in African participants with hypertension provides strong evidence for the devastating outcomes of untreated hypertension as it reports a 17.8% pooled prevalence of CKD in people with hypertension.[43] In the total sample of men in the current study the most prevalent combination of three diseases was dyslipidaemia, hypertension and kidney disease (2% of the total sample) while in the women this was dyslipidaemia, hypertension and HIV (3% of the total sample). The co-occurrence of kidney disease and hypertension was more common than kidney disease and diabetes in both men and women, a finding which has previously been reported in Africa.[44] Several multimorbidity patterns including a 'cardiorespiratory' pattern and a 'metabolic' pattern were identified across countries in the Study on global AGEing and adult health (SAGE) reporting data on 12 chronic conditions, although the factors associated with multimorbidity differed between countries and could be explained by their diverse development status.[11] Reporting on the South African data only, Chidumwa *et al*, identified three groups using latent class analysis.[7] While 88% of the sample were classified as minimal multimorbidity risk, 11% were classified as concordant (hypertension and diabetes) multimorbidity and 6% as discordant (angina, asthma, chronic lung disease, arthritis and depression) multimorbidity.[7]

Identifying factors associated with multimorbidity clustering is important in designing country or region-specific strategies to manage multimorbidity. Results from this study as well as global and African studies have reported an increased risk of multimorbidity with increasing age.[10 30] Although this association is well accepted and understood, in the study by Garin *et al*,[11] multimorbidity prevalence decreased with age in the South African cohort, which they suggested may be due to the decrease in HIV prevalence with age. The current study also reported an 11%–14% higher risk of multimorbidity with each 1 kg/m$^2$ increase in BMI in both men and women. Overweight and obesity are well known risk factors for NCDs such as diabetes, hypertension and dyslipidaemia, particularly a low HDL-C, and obesity is a targeted risk factor by the WHO in their Global Action Plan for the prevention and control of NCDs.[45] Although age and BMI were associated with multimorbidity prevalence in both men and women, lifestyle correlates displayed sex differences. In the women being a former consumer of alcohol was associated with nearly two times the risk of having multimorbidity, while men who were current, non-problematic consumers of alcohol were at lower risk of multimorbidity. Data from the same SSA cohort has reported that current alcohol consumption is lower in women compared with men at all AWI-Gen sites, with the characteristics of alcohol consumption such as type and frequency of consumption being significantly different between the sexes.[23] This may explain the different

associations with multimorbidity risk in the current study, and together with the finding that time spent in MVPA was inversely associated with multimorbidity risk in men only, suggests that further intervention studies to reduce multimorbidity risk may need to be stratified by sex. The influence of socioeconomic patterns on disease risk is well recognised, and may differ between countries at different stages of the epidemiological transition.[27] Employment was associated with a reduced risk of multimorbidity in this study and supports the inverse association between SES and disease risk reported by others in LMICs.[46] We have shown that when compared with men who have never married or been in a union, men who are and men who have previously been in a union (divorced, separated, partner deceased) were at a significantly higher risk of multimorbidity. In their study of adults living in four SSA countries, Ajayi *et al*[47] reported similar associations with BMI but in only two of the sites, rural and periurban Uganda, with no associations in adults from Tanzania and South Africa. We have previously shown in the AWI-Gen cohort that married men had a higher BMI compared with their unmarried counterparts, which consisted of those who had never been married or were no longer married,[48] while results from the Prospective Urban and Rural Epidemiological (PURE) study have shown that people who have experienced marital loss are at higher CVD risk.[49] Marital status may represent social position, SES as well as cultural beliefs around body size and marriage, and these results highlight the complexity of the association with disease risk.

It is acknowledged that these data are cross-sectional and single screen testing without follow-up may result in the overestimation of diseases such as CKD. A further limitation of the study is that data were only available for four of the seven conditions for the Soweto women; however, description of the disease profile of this subsample still makes a contribution to the limited literature from LMICs. Further, only prevalence and correlates of multimorbidity could be identified; however, the sample size is large and represents four SSA countries to help understand differences and similarities between populations at different stages of the epidemiological transition. These prevalence figures make an important contribution to the increasing knowledge around multimorbidity and may be particularly useful in developing indices that can be used as part of machine learning approaches in predicting multimorbidity in the future.

In conclusion, this study has shown that nearly half of the population-based sample of women and more than a third of the men between the ages of 40 and 60 years from South, East and West Africa are living with two or more chronic diseases. Common disease clusters have been identified and future efforts should focus on managing multiple commonly occurring diseases rather than single diseases. Although the prevalence of multimorbidity is higher in women living in South Africa, it is expected that as the obesity epidemic continues to increase that the prevalence will increase in men as well as East and West

Africa. Identifying correlates of multimorbidity is critical to providing focused and effective interventions.

**Author affiliations**
[1]SAMRC/Wits Developmental Pathways for Health Research Unit, Department of Paediatrics, Faculty of Health Sciences, University of the Witwatersrand, Johannesburg, South Africa
[2]Department of Biochemistry and Forensic Sciences, School of Chemical and Biochemical Sciences, C.K. Tedam University of Technology and Applied Sciences, Navrongo, Ghana
[3]Navrongo Health Research Centre, Ghana Health Service, Accra, Ghana
[4]African Population and Health Research Center, Nairobi, Kenya
[5]Clinical Research Unit of Nanoro, Institut de Recherche en Sciences de la Santé, Ouagadougou, Burkina Faso
[6]Sydney Brenner Institute of Molecular Bioscience, University of the Witwatersrand, Johannesburg, South Africa
[7]Department of Public Health, University of Limpopo, Sovenga, South Africa
[8]Department of Chemical Pathology, National Health Laboratory Service, Faculty of Health Sciences, University of the Witwatersrand, Johannesburg, South Africa
[9]Wits Donald Gordon Medical Centre, University of the Witwatersrand, Johannesburg, South Africa
[10]South African Medical Research Council/Wits Rural Public Health and Health Transitions Research Unit, School of Public Health, Faculty of Health Sciences, University of the Witwatersrand, Johannesburg, South Africa
[11]Health and Systems for Health, African Population and Health Research Center, Nairobi, Kenya
[12]Julius Global Health, Julius Centre for Health Sciences and Primary Care, Utrecht University, Utrecht, The Netherlands
[13]Department of Medicine, Faculty of Health Sciences, University of the Witwatersrand, Johannesburg, South Africa
[14]Institut de Recherche en Sciences de la Santé, Ouagadougou, Burkina Faso
[15]Sydney Brenner Institute for Molecular Bioscience, Faculty of Health Sciences, University of the Witwatersrand, Johannesburg, South Africa
[16]School of Human Development and Health, University of Southampton, Southampton, UK

**Acknowledgements** This study would not have been possible without the generosity of the participants who spent many hours responding to questionnaires, being measured and having samples taken. We wish to acknowledge the contributions of our field workers, phlebotomists, laboratory scientists, administrators, data personnel and other investigators who contributed to the data and sample collections, processing, storage and shipping. Investigators responsible for the conception and design of the AWI-Gen study include the following Michèle Ramsay (PI, Wits), Osman Sankoh (co-PI, INDEPTH), Stephen Tollman and Kathleen Kahn (Agincourt PI), Marianne Alberts (Dikgale (renamed DIMAMO) PI (deceased)), Catherine Kyobutungi (Nairobi PI), Halidou Tinto (Nanoro PI), Abraham Oduro (Navrongo PI), Shane Norris (Soweto PI), and Scott Hazelhurst, Nigel Crowther, Himla Soodyall and Zane Lombard (Wits). We would like to acknowledge each of the following investigators for the significant contributions to this research, mentioned according to affiliation are: Wits AWI-Gen Collaborative Centre—Stuart Ali, Ananyo Choudhury, Scott Hazelhurst, Freedom Mukomana, Cassandra Soo; Soweto (DPHRU): Nomses Baloyi, Yusuf Guman.

**Contributors** LKM, SAN and MR conceptualised the study. RM did the analyses and prepared all the tables and figures. LKM wrote the first manuscript draft and RM, SAN and MR offered further guidance on the analyses, interpretation and writing. All authors reviewed the manuscript and approved the manuscript before submission. MR takes full responsibility for the work and/or the conduct of the study, has access to the data, and controlled the decision to publish.

**Funding** The AWI-Gen Collaborative Centre is funded by the National Human Genome Research Institute (NHGRI), the National Institute of Environmental Health Sciences (NIEHS), of the National Institutes of Health (NIH) under award number U54HG006938, as part of the H3Africa Consortium, and by the Department of Science and Innovation, South Africa, award number DST/CON 0056/2014. MR is a South African Research Chair in Genomics and Bioinformatics of African populations hosted by the University of the Witwatersrand, funded by the Department of Science and Technology and administered by National Research Foundation of South

Africa (NRF). The Birth to Twenty Cohort (Soweto, South Africa) is supported by University of the Witwatersrand, the South African Medical Research Council, and The Wellcome Trust, UK. SAN is supported by the DST/NRF Centre of Excellence in Human Development at the University of the Witwatersrand, Johannesburg. ANW is supported by the Fogarty International Centre, National Institutes of Health (grant number K43TW010698). This paper describes the views of the authors and does not necessarily represent the official views of the National Institutes of Health or the National Research Foundation of South Africa who funded this research.

**Competing interests** None declared.

**Patient and public involvement** Patients and/or the public were not involved in the design, or conduct, or reporting, or dissemination plans of this research.

**Patient consent for publication** Not applicable.

**Ethics approval** This study involves human participants and was approved by Human Research Ethics Committee (Medical) of the University of the Witwatersrand (certificate numbers M121029 and M170880). Each study centre obtained local ethics approval. Participants gave informed consent to participate in the study before taking part.

**Provenance and peer review** Not commissioned; externally peer reviewed.

**Data availability statement** Data are available upon reasonable request. The data have been submitted to the European Genome-Phenome Archive (EGA), accession number EGA00001002482. The Human Heredity and Health in Africa (H3Africa) Data and Biospecimen Access Committee (DBAC) will review requests for the AWI-Gen phenotype dataset. Related documents including study protocol and statistical analysis plan will be available upon request from the corresponding author.

**ORCID iDs**
Lisa K Micklesfield http://orcid.org/0000-0002-4994-0779
Godfred Agongo http://orcid.org/0000-0002-4218-5424
Palwende Boua http://orcid.org/0000-0001-8325-2665
Solomon SR Choma http://orcid.org/0000-0003-1338-6523
Eric Maimela http://orcid.org/0000-0003-2843-4663
Shukri F Mohamed http://orcid.org/0000-0002-8693-1943
Alisha N Wade http://orcid.org/0000-0002-1158-2523
Shane A Norris http://orcid.org/0000-0001-7124-3788

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
