## [Reviewer comments · BMJ Open]

ARTICLE DETAILS

TITLE (PROVISIONAL)	Identifying the prevalence and correlates of multimorbidity in middle-aged men and women: a cross-sectional population-based study in four African countries
AUTHORS	Micklesfield, Lisa; Munthali, Richard; Agongo, Godfred; Asiki, G; Boua, Palwende; Choma, Solomon SR; Crowther, Nigel; Fabian, June; Gómez-Olivé, F. Xavier; Kabudula, Chodziwadziwa; Maimela, Eric; Mohamed, Shukri; Nonterah, Engelbert; Raal, Frederick; Sorgho, Hermann; Tluway, Furahini; Wade, Alisha N.; Norris, Shane; Ramsay, Michele

VERSION 1 – REVIEW

REVIEWER	Rai, Pragya West Virginia University School of Pharmacy
REVIEW RETURNED	20-Sep-2022

GENERAL COMMENTS	Thank you for the opportunity to review the paper. This is a well-written and articulated paper. Some comments for your consideration. Methods Section Socio-demographic and anthropometric data: How was current problematic and non-problematic in alcohol consumption defined? Was this based on what the individual considered problematic or were they provided instructions in the survey (e.g., bottles/day) to define this? Can add this to this paragraph for clarification. Multimorbidity: Was there a reason only these 7 conditions were selected and not those from Charlson Comorbidity Index or the list of chronic conditions developed by the Multiple Chronic Conditions working group (doi:10.5888/pcd10.120239)? Why was data on only 4 conditions for women from Soweto was available, especially since this was self-reported data? Within para for each individual condition, not all conditions have the statement about data availability for Soweto women. Please be consistent- either remove the statement from all individual disease paragraph (as it is already mentioned in the first para) or add it for all conditions. How are you categorizing multimorbidity? Please see next para for the reason for this question. Statistical analysis: Under multimorbidity, it is defined as presence of 2+ conditions. Then should the outcome not be multimorbidity (yes/no), where you categorize yes if 2+ conditions are present and no if 0-1 conditions are present? The way it is currently explained (Since the multimorbidity outcome had three categories, zero, one, and at least two conditions.....), you are categorizing "comorbidity" rather than "multimorbidity". I would suggest to either
---

	define comorbidity (with a focus on multimorbidity) or reclassify the variable multimorbidity and redo the analysis. Patient and public involvement: I don't understand this statement. Are the individuals who were surveyed not part of public? Or patients are the various institutes? Results Socio-demographic and lifestyle factors: It is mentioned that alcohol status is unknown among Soweto women. Please mention this in Methods section as well.
--	--

REVIEWER	Vargese, Saritha
REVIEW RETURNED	06-Oct-2022

GENERAL COMMENTS	Multimorbidity is a relevant topic especially with epidemiological and demographic transition. I appreciate you taking up this important work. However there are several areas which has to be revised and improved before the paper can be considered for publication. In the abstract, the conclusion needs to be revised as it is not coherent. In the background, the information is disconnected, spread-out and doesn't serve to build a strong study rationale. There is need to revise and modify the introduction section. There are some methodological limitations which needs to be addressed. It has been marked in the manuscript. There are formatting and grammar mistakes all through the text which needs to be addressed. Some of them have been marked in the text. The strengths and limitations need to be detailed further. Several other comments have been added in the manuscript.
---

VERSION 1 – AUTHOR RESPONSE

Reviewer: 1

Dr. Pragya Rai, West Virginia University School of Pharmacy

Comments to the Author:

Thank you for the opportunity to review the paper. This is a well-written and articulated paper.

Some comments for your consideration.

Methods Section

Socio-demographic and anthropometric data: How was current problematic and non-problematic in alcohol consumption defined? Was this based on what the individual considered problematic or were they provided instructions in the survey (e.g., bottles/day) to define this? Can add this to this paragraph for clarification.

- Thank you for this suggestion. Problematic drinking of alcohol was determined by the CAGE questionnaire. We have now included the reference in the manuscript.

Multimorbidity: Was there a reason only these 7 conditions were selected and not those from Charlson Comorbidity Index or the list of chronic conditions developed by the Multiple Chronic Conditions working group (doi:10.5888/pcd10.120239)? Why was data on only 4 conditions for

women from Soweto was available, especially since this was self-reported data? Within para for each individual condition, not all conditions have the statement about data availability for Soweto women. Please be consistent- either remove the statement from all individual disease paragraph (as it is already mentioned in the first para) or add it for all conditions. How are you categorizing multimorbidity? Please see next para for the reason for this question.

- The seven conditions included in the definition of multimorbidity were selected based on the data that was available in the study. The study was not originally designed to measure multimorbidity so we were only able to use the data that was available. Data was only available for four conditions for the Soweto women as they were recruited from an existing cohort on whom some of the data had already been collected. We have referenced the protocol paper which details this recruitment.

- We have removed the statement about the Soweto women from the individual disease paragraphs.

- Multimorbidity was categorised as having two or more of the seven conditions. This definition of multimorbidity has been adopted by the World Health Organisation.

Statistical analysis: Under multimorbidity, it is defined as presence of 2+ conditions. Then should the outcome not be multimorbidity (yes/no), where you categorize yes if 2+ conditions are present and no if 0-1 conditions are present? The way it is currently explained (Since the multimorbidity outcome had three categories, zero, one, and at least two conditions), you are categorizing "comorbidity" rather than "multimorbidity". I would suggest to either define comorbidity (with a focus on multimorbidity) or reclassify the variable multimorbidity and redo the analysis.

- Within the Statistical analysis section we now describe the outcome as follows:

“Since the outcome had three categories, zero conditions, one condition, and at least two conditions (multimorbidity) of the seven conditions under study, multinomial logistic regression was used to explore the factors associated with either one condition or at least two conditions, with none as the reference group, in men and women separately.” Factors associated with having one condition and with having two or more conditions (multimorbidity) are clearly described in the results section (table and text).

Patient and public involvement: I don't understand this statement. Are the individuals who were surveyed not part of public? Or patients are the various institutes?

- Page 10, line 14: This is a new requirement for submission to BMJ Open to describe whether patients or the public were involved in the research process ie. design and conduct of the study, and dissemination of the results. Although we have actively engaged with the community with regard to dissemination of their results they have not been involved, to date, with the design of the study. For this reason we have stated that 'This study did not involve any patients or the public', as requested by the journal.

Results

Socio-demographic and lifestyle factors: It is mentioned that alcohol status is unknown among Soweto women. Please mention this in Methods section as well.

- Thank you, we have added this to the Methods section.

Reviewer: 2

Saritha Vargese

Comments to the Author:

Dear authors,

Multimorbidity is a relevant topic especially with epidemiological and demographic transition. I appreciate you taking up this important work. However there are several areas which has to be revised and improved before the paper can be considered for publication.

Thank you. Please see responses below and as track changes in the marked version of the original manuscript.

In the abstract, the conclusion needs to be revised as it is not coherent.

The conclusion has been revised to make it more coherent and now reads as follows:
“The high prevalence of multimorbidity in middle-aged adults in sub-Saharan Africa is of concern, with women currently at higher risk. This prevalence is expected to increase in men, as well as in the East and West African region with the ongoing epidemiological transition. Identifying common disease clusters and correlates of multimorbidity is critical to providing effective interventions.”

In the background, the information is disconnected, spread-out and doesn't serve to build a strong study rationale. There is need to revise and modify the introduction section.

Thank-you for this suggestion. The introduction has been modified and changes have been tracked in the manuscript.

There are some methodological limitations which needs to be addressed. It has been marked in the manuscript.

Please see responses to more specific comments below:

- Page 7, line 56: the seven conditions included within the definition of multimorbidity were selected based on the data that was available in the study. The study was not originally designed to measure multimorbidity so we were only able to use the data that was available.

- Page 8, line 11-13: HIV is extremely low in West Africa and due to the stigma associated with people living with HIV it was decided not to offer HIV tests. We accept not being able to present HIV prevalence in all sites is a limitation but still feel that the prevalence of multimorbidity including the other diseases is still of interest.

- The references for the hypertension and diabetes definitions have been included.

- Page 9, line 12: We acknowledge that stroke and TIA could be included as cerebrospinal disease but due to the low numbers who had experienced either of these conditions we felt that it would be better to include under the cardiovascular disease condition. We made sure to clearly explain this in the methods.

- Page 9, line 23: We acknowledge that some of the methods were different between the sites, with some data missing from the Soweto women, however due to the large sample size and the valuable contribution of this data to the limited literature we felt that it would be preferable to present the data as it is, even with the limitations that are well described.

- Page 10, line 14: This is a new requirement for submission to BMJ Open to describe whether patients or the public were involved in the research process ie. design and conduct of the study, and dissemination of the results. Although we have actively engaged with the community with regard to dissemination of their results they have not been involved, to date, with the design of the study. For this reason we have stated that 'This study did not involve any patients or the public', as requested by the journal.

There are formatting and grammar mistakes all through the text which needs to be addressed. Some of them have been marked in the text.

These have been addressed throughout the manuscript. Please see below for responses to specific comments in the rest of the manuscript:

- Page 13, line 5: we disagree that Soweto women should be excluded from general analyses. We acknowledge that it is a limitation, and have now included this in the limitations section, but we feel that these data still contribute to the findings of the study. We have described the Soweto women's sample well including what diseases we were able to quantify and believe that the reader will be able to interpret the findings accordingly.

- Discussion: we have removed some of the results from the first paragraph of the Discussion but feel that the geographical prevalence results are still relevant here to highlight the differences between south, east and west Africa, and between men and women.

- The sex differences in multimorbidity prevalence in the current study are consistent with other studies, which we have now highlighted in the discussion. This is thought to be due to biological, socio-cultural or environmental influences.

The strengths and limitations need to be detailed further.

- More detail has been included in this paragraph acknowledging the limited data available on the Soweto women.

Several other comments have been added in the manuscript.

- These have been addressed throughout the manuscript. Although the reviewer mentions in the concluding paragraph that referral to the obesity epidemic should be removed as the study did not measure obesity, we feel that this should remain as we did include data in the Introduction highlighting the high prevalence of obesity in South Africa.

VERSION 2 – REVIEW

REVIEWER	Rai, Pragya West Virginia University School of Pharmacy
REVIEW RETURNED	23-Dec-2022
GENERAL COMMENTS	No further comments

VERSION 2 – AUTHOR RESPONSE